# TRAINING RNNs AS FAST AS CNNs

## ABSTRACT

Common recurrent neural network architectures scale poorly due to the intrinsic difficulty in parallelizing their state computations. In this work, we propose the Simple Recurrent Unit (SRU) architecture, a recurrent unit that simplifies the computation and exposes more parallelism. In SRU, the majority of computation for each step is independent of the recurrence and can be easily parallelized. SRU is as fast as a convolutional layer and 5-10x faster than an optimized LSTM implementation. We study SRUs on a wide range of applications, including classification, question answering, language modeling, translation and speech recognition. Our experiments demonstrate the effectiveness of SRU and the trade-off it enables between speed and performance. We open source our implementation in PyTorch and CNTK.

## 1 INTRODUCTION

Recurrent neural networks (RNN) are at the core of state-of-the-art approaches for a large number of natural language tasks, including machine translation (Cho et al., 2014; Bahdanau et al., 2015; Jean et al., 2015; Luong et al., 2015), language modeling (Zaremba et al., 2014; Gal & Ghahramani, 2016; Zoph & Le, 2016), opinion mining (Irsoy & Cardie, 2014), situated language understanding (Mei et al., 2016; Misra et al., 2017), and question answering (Seo et al., 2016a; Chen et al., 2017). Key to many of these advancements are architectures of increasing capacity. However, these networks are difficult to scale. During learning, the sequential dependencies that are central to recurrent architectures limit parallelization potential. This results in slow development and makes rigorous parameter tuning intractable. Similar problems occur during deployment when slow inference creates challenges for real-time systems at scale. In this paper, we describe the Simple Recurrent Unit (SRU), a recurrent architecture that balances serial and parallelized computation. We evaluate SRU and show the speed gains it provides generalize across a set of core tasks, while maintaining and even improving overall performance over common architectures.

Recurrent networks process sequences of symbols (e.g., words in a sentence) one symbol at a time. In commonly used architectures, including Long Short-term Memory (LSTM; Hochreiter & Schmidhuber, 1997) and Gated Recurrent Units (GRU; Cho et al., 2014), the computation in each step depends on completing the previous step. As a result, in contrast to operations such as convolution and attention, recurrent computations are less amenable to parallelization. We propose to do the majority of the computation for each step without depending on completing previous computations, which allows for to easily parallelize it. The result of this computation are then combined via a fast recurrent structure. Figure 1 illustrates the difference between the approaches.

While even a naive implementation of our approach leads to improvements in performance, one of its key advantage is enabling optimization particularly fitting to existing hardware architectures. Removing the dependencies between time steps for the most expensive operations allows to parallelize across different dimensions and time steps. We also perform a CUDA-level optimization by compiling element-wise operations of the computations into a single kernel function call. Figure 2 compares our architecture's runtimes to common architectures.

We experiment with a diverse set of core problems to evaluate our architecture, including text classification, question answering, language modeling, machine translation, and speech recognition. Our approach is competitive and even outperforms common recurrent and convolutional architectures, while delivering significant speedups. We also study the relation between speed and performance, and show SRU provides fine-grained control of the tradeoff between the two.

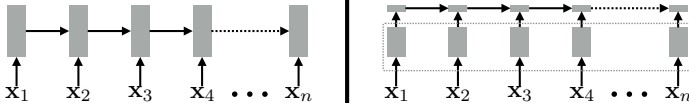

Figure 1: Illustration of the difference between common RNN architectures (left) and our approach (right). In common architectures, the entire computation (gray block) for each step $\mathbf{x}_t$, $t = 1, \ldots, n$ depends on completing the previous step. This impedes any parallelization between steps. In contrast, we propose to process the input at each step independently of the other inputs (larger gray block) and do the recurrent combination with relatively lightweight computation (small gray block). The majority of the computation (surrounded by the dashed line) can then be easily parallelized.

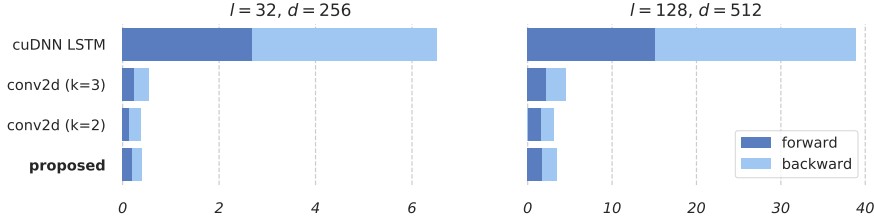

Figure 2: Average processing time (in milliseconds) of a batch of 32 samples using `cuDNN LSTM`, word-level convolution `conv2d`, and the proposed SRU. $l$ number of tokens per sequence, $d$: feature dimension and $k$: feature width. See Section 4 for details of the setup used.

## 2  METHOD

### 2.1  SIMPLE RECURRENT UNITS

Most recurrent architectures, including LSTM and GRU, use *gating* to control the information flow to alleviate vanishing and exploding gradient problems. We define a gate to be composed of a single-layer feed-forward network with a sigmoid activation. The gate output is used in a point-wise multiplication operation to combine two inputs, for example the current and previous time stamps. The computation of the feed-forward network, especially the matrix multiplication, is the most expensive operation in this process, while the point-wise multiplication is relatively lightweight. The key design decision in SRU is making the gate computation dependent only on the current input of the recurrence. This leaves only the point-wise multiplication computation as dependent on previous steps. The matrix multiplications involved in the feed-forward network can then be easily parallelized.

The basic form of SRU includes a single *forget gate*. Given an input $\mathbf{x}_t$ at time $t$, we compute a linear transformation $\tilde{\mathbf{x}}_t$ (Lei et al., 2017; Lee et al., 2017) and the forget gate $\mathbf{f}_t$:

$$\tilde{\mathbf{x}}_t = \mathbf{W}\mathbf{x}_t$$
$$\mathbf{f}_t = \sigma(\mathbf{W}_f \mathbf{x}_t + \mathbf{b}_f) \ .$$

This computation depends on $\mathbf{x}_t$ only, which enables computing it in parallel across all time steps. The forget gate is used to modulate the internal state $\mathbf{c}_t$, which is used to compute the output state $\mathbf{h}_t$:

$$\mathbf{c}_t = \mathbf{f}_t \odot \mathbf{c}_{t-1} + (1 - \mathbf{f}_t) \odot \tilde{\mathbf{x}}_t$$
$$\mathbf{h}_t = g(\mathbf{c}_t) \ ,$$

where $g(\cdot)$ is an activation function used to produce the output state $\mathbf{h}_t$.

The complete architecture also includes skip connections, which have been shown to improve training of deep networks with a large number of layers (He et al., 2016; Srivastava et al., 2015; Wu et al., 2016a). We use highway connections (Srivastava et al., 2015), and add a *reset gate* $\mathbf{r}_t$ computed similar to the forget gate $\mathbf{f}_t$. The reset gate is used to compute the output state $\mathbf{h}_t$ as a combination of

the internal state $g(\mathbf{c}_t)$ and the input $\mathbf{x}_t$. The complete architecture is:

$$\tilde{\mathbf{x}}_t = \mathbf{W}\mathbf{x}_t \tag{1}$$

$$\mathbf{f}_t = \sigma(\mathbf{W}_f\mathbf{x}_t + \mathbf{b}_f) \tag{2}$$

$$\mathbf{r}_t = \sigma(\mathbf{W}_r\mathbf{x}_t + \mathbf{b}_r) \tag{3}$$

$$\mathbf{c}_t = \mathbf{f}_t \odot \mathbf{c}_{t-1} + (1 - \mathbf{f}_t) \odot \tilde{\mathbf{x}}_t \tag{4}$$

$$\mathbf{h}_t = \mathbf{r}_t \odot g(\mathbf{c}_t) + (1 - \mathbf{r}_t) \odot \mathbf{x}_t \tag{5}$$

## 2.2 RELATION TO COMMON ARCHITECTURES

Existing RNN architectures use the previous output state $\mathbf{h}_{t-1}$ in the recurrence computation. For example, in LSTM, the forget gate vector is computed by $\mathbf{f}_t = \sigma(\mathbf{W}_f\mathbf{x}_t + \mathbf{R}_f\mathbf{h}_{t-1} + \mathbf{b}_f)$. Including $\mathbf{R}\mathbf{h}_{t-1}$ breaks independence and parallelization: each dimension of the hidden state $\mathbf{h}_t$ depends on $\mathbf{h}_{t-1}$, and the computation of $\mathbf{h}_t$ has to wait until $\mathbf{h}_{t-1}$ is fully computed. Similar design choices are present in GRU and other RNN variants, where $\mathbf{h}_{t-1}$ is used throughout the computation.

We propose to completely drop the connection between the gating computations of step $t$ and the states of step $t - 1$, $\mathbf{h}_{t-1}$ and $\mathbf{c}_{t-1}$. Given a sequence of input vectors $\{\mathbf{x}_1, \cdots, \mathbf{x}_n\}$, $\{\tilde{\mathbf{x}}_t, \mathbf{f}_t, \mathbf{r}_t\}$ for different $t = 1 \cdots n$ are independent and can be computed in parallel. The computation bottleneck of our architecture is the three matrix multiplications in Equations 1-3. After computing $\tilde{\mathbf{x}}_t$, $\mathbf{f}_t$ and $\mathbf{r}_t$, Equations 4 and 5, where all operations are element-wise, are fast to compute.

## 2.3 CUDA-LEVEL OPTIMIZATION

Optimizing SRU is similar to how LSTM is optimized in `cuDNN LSTM` (Appleyard et al., 2016). The SRU formulation permits two optimizations. First, matrix multiplications across all time steps can be batched, which significantly improves the computation intensity and GPU utilization. Grouping the matrix multiplications in Equations 1-3 into a single batch is formulated as

$$\mathbf{U}^\top = \left( \begin{array}{c} \mathbf{W} \\ \mathbf{W}_f \\ \mathbf{W}_r \end{array} \right) [\mathbf{x}_1, \mathbf{x}_2, \cdots, \mathbf{x}_n] \ ,$$

where $n$ is the sequence length, $\mathbf{U} \in \mathbb{R}^{n \times 3d}$ is the resulting matrix, and $d$ is the hidden state size. When the input is a mini-batch of $k$ sequences, $\mathbf{U}$ would be a tensor of size $(n, k, 3d)$. Second, all element-wise operations of the sequence can be fused into one kernel function and parallelized across the dimensionality of the hidden state $d$. Without the fusion, operations such as addition $+$ and sigmoid activation $\sigma()$ would each invoke a separate function call, and incur additional kernel launching latency and data moving costs. Algorithm 1 shows the pseudocode of the fused kernel function. The implementation of a bidirectional SRU is similar: the matrix multiplications of both directions are batched, and a fused kernel is created to handle and parallelize both directions.

## 3 RELATED WORK

Improving on common architectures for sequence processing has recently received significant attention (Greff et al., 2015; Balduzzi & Ghifary, 2016; Miao et al., 2016; Zoph & Le, 2016; Lee et al., 2017; Vaswani et al., 2017). Our approach is closely related to recent work on recurrent convolutions (RCNN; Lei et al., 2015; 2016), kernel networks (KNN; Lei et al., 2017), and Quasi-RNN (Bradbury et al., 2017). Both RCNN and Quasi-RNN incorporate word-level convolutions into recurrent unit with sequential gated pooling. KNN generalizes RCNN and provides a theoretical view by linking the model class to sequence kernels. SRU can be viewed as a simplified version, or a special case, of RCNN, KNN, and Quasi-RNN, where the window size is set to 1 and highway connections Srivastava et al. (2015) are added to facilitate increased network depth. We discuss the relation of SRU to Quasi-RNN in more detail and evaluate the effect of the differences in Appendix A.

Various strategies have been proposed to scale network training (Goyal et al., 2017) or specifically to speed up recurrent networks (Diamos et al., 2016; Kuchaiev & Ginsburg, 2017; Shazeer et al., 2017). Our CUDA-level optimization for SRU is inspired by `cuDNN LSTM` (Appleyard et al., 2016). While `cuDNN LSTM` requires six optimization steps, SRU only requires two optimizations to produce

---

**Algorithm 1** Mini-batch version of the forward pass defined in Equations 1-5.

---

**Indices:** Sequence length $n$, $l = 1, \cdots, n$; mini-batch size $k$, $i = 1, \cdots, k$; hidden state dimension $d$, $j = 1, \cdots, d$; and $j' = 1, \cdots, 3d$.

**Input:** Input sequences batch $\mathbf{x}[l, i, j]$; grouped matrix multiplication result $\mathbf{U}[l, i, j']$; bias terms $\mathbf{b}_f[j]$ and $\mathbf{b}_r[j]$; and initial state $\mathbf{c}_0[i, j]$.

**Output:** Output $\mathbf{h}[\cdot, \cdot, \cdot]$ and internal $\mathbf{c}[\cdot, \cdot, \cdot]$ states.

  Initialize $\mathbf{h}[\cdot, \cdot, \cdot]$ and $\mathbf{c}[\cdot, \cdot, \cdot]$ as two $n \times k \times d$ tensors.

  **for** $i = 1, \cdots, k; j = 1, \cdots, d$ **do**     // Parallelize over $i$ and $j$

    $\mathbf{c}' = \mathbf{c}_0[i, j]$

    **for** $l = 1, \cdots, n$ **do**

      $f = \sigma\left(\mathbf{U}[l, i, j + d] + \mathbf{b}_f[j]\right)$     // Forget gate

      $r = \sigma\left(\mathbf{U}[l, i, j + d \times 2] + \mathbf{b}_r[j]\right)$     // Reset gate

      $c = f \times c + (1 - f) \times \mathbf{U}[l, i, j]$     // Current internal state

      $h = f \times g(c) + (1 - r) \times \mathbf{x}[l, i, j]$     // Current output state

      $\mathbf{c}[l, i, j] = c$

      $\mathbf{h}[l, i, j] = h$

  **return** $\mathbf{h}[\cdot, \cdot, \cdot]$ and $\mathbf{c}[\cdot, \cdot, \cdot]$

---

significant speed-up. The convolution-based Quasi-RNN architecture (Bradbury et al., 2017) uses similar CUDA-level optimizations such as `conv2d` operation or batched matrix multiplications. The topic of improving learning times was also studied. For example, Goyal et al. (2017) addressed stability issues of distributed training with large mini-batches to improve training time. Our approach can be combined with such training procedures.

The design of simple recurrent architectures, such as SRU and other related architectures, raises questions about representational power. Balduzzi & Ghifary (2016) applies type-preserving transformations to the discuss the capacity of various simplified RNN architectures. Recent work has demonstrated the connection between neural networks and kernels (Anselmi et al., 2015; Daniely et al., 2016; Zhang et al., 2016). In particular, Lei et al. (2017) shows that a broad model class, including SRU and word-level CNN, can be seen as embedding sequence similarity functions, such as string kernels (Lodhi et al., 2002), into a hidden space. Layer stacking can then be interpreted as using higher-order sequence similarities, which introduces more non-linearity and representational power. We empirically show SRU can achieve compelling results by stacking multiple layers.

## 4 EXPERIMENTS

We evaluate SRU with text classification, question answering, language modeling, machine translation, and speech recognition tasks. This set of tasks provides broad coverage of application and computation challenges. Training time on these benchmarks ranges from minutes (classification) to days (speech).

Unless noted otherwise, timing experiments are performed on PyTorch and a desktop machine with a single Nvidia GeForce GTX 1070 GPU, Intel Core i7-7700K Processor, CUDA 8 and cuDNN 6021. We use variational dropout (Gal & Ghahramani, 2016) in addition to the standard dropout for RNN regularization. We set $g(\cdot) = \texttt{tanh}$ for all our experiments, unless specified otherwise.

The main question we study is the performance-speed trade-off SRU provides in comparison to other recurrent architectures. We stack multiple layers of SRU to directly substitute other recurrent or convolutional modules. We minimize hyper-parameter tuning and architecture engineering for a fair comparison. Such efforts have a non-trivial impact on the results, which are beyond the scope of our experiments. As much as possible, the model configurations are identical to prior work.

### 4.1 CLASSIFICATION

**Dataset**   We use six classification tasks from Kim (2014):[1] movie review sentiment (MR; Pang & Lee, 2005), subjectivity (SUBJ; Pang & Lee, 2004), customer reviews polarity (CR; Hu & Liu, 2004), TREC question type (TREC; Li & Roth, 2002), MPQA opinion polarity (MPQA; Wiebe et al.,

---

[1] https://github.com/harvardnlp/sent-conv-torch

| Model | CR | SUBJ | MR | TREC | MPQA | SST |
|---|---|---|---|---|---|---|
| CNN (Kim, 2014) | 82.2 ±2.2 | 92.9 ±0.7 | 79.1 ±1.5 | 93.2 ±0.5 | 88.8 ±1.2 | 85.3 ±0.4 |
| LSTM | 82.7 ±2.9 | 92.4 ±0.6 | 80.3 ±1.5 | 93.1 ±0.9 | 89.2 ±1.0 | 87.9 ±0.6 |
| SRU | 84.8 ±1.3 | 93.4 ±0.8 | 82.2 ±0.9 | 93.9 ±0.6 | 89.7 ±1.1 | 89.1 ±0.3 |

Table 1: Classification (Section 4.1) test accuracies on six benchmarks.

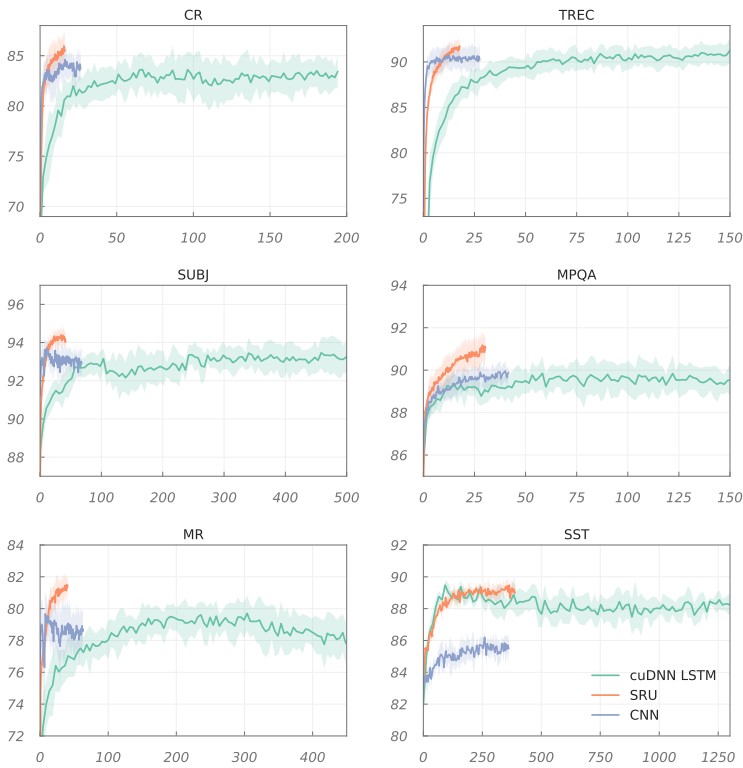

Table 2: Mean validation accuracies (y-axis) of LSTM, CNN, and SRU for the first 100 epochs on the six classification benchmarks. X-axis: training time used (in seconds).

2005), and the Stanford sentiment treebank (SST; Socher et al., 2013).[2] Following (Kim, 2014), we use `word2vec` embeddings trained on 100 billion Google News tokens. Word embeddings are normalized to unit vectors and are fixed during training.

**Setup** We train RNN encoders and use the last output state to predict the class label for a given sentence. We use a two-layer RNN encoder with 128 hidden dimensions. For SST, which provides more data, we use a four-layer RNN. We also compare to the CNN model of Kim (2014), with the same filter windows of 3, 4, and 5 as the original work. We use `Adam` (Kingma & Ba, 2014) with default 0.001 learning rate and 0 weight decay. We train for 100 epochs, and perform 10-fold cross validation when no standard split is specified. The result on SST is averaged over five independent trials. We tune dropout probability among {0.1, 0.3, 0.5, 0.7} and report the best results.

**Results** Table 1 presents test results on the six benchmarks. Our model consistently outperforms the other models across the datasets. Figure 2 shows validation performance relative to training time for SRU, `cuDNN LSTM`, and the CNN model. Our SRU implementation is significantly faster than `cuDNN LSTM`. For example, on the movie review task (MR), our model completes 100 training epochs within 40 seconds, while `cuDNN LSTM` takes more than 450 seconds.

---

[2]We use the binary version of the Stanford sentiment treebank.

| Model | # layers | Dim. | Size | Dev EM | Dev F1 | Time per epoch RNN | Total |
|---|---|---|---|---|---|---|---|
| Chen et al. (2017) | 3 | 128 | 4.1m | 69.5 | 78.8 | - | - |
| Bi-LSTM | 3 | 128 | 4.1m | 69.6 | 78.7 | 534s | 670s |
| Bi-LSTM | 4 | 128 | 5.8m | 69.6 | 78.9 | 729s | 872s |
| Bi-SRU | 3 | 128 | 2.0m | 69.1 | 78.4 | 60s | 179s |
| Bi-SRU | 4 | 128 | 2.4m | 69.7 | 79.1 | 74s | 193s |
| Bi-SRU | 5 | 128 | 2.8m | 70.3 | 79.5 | 88s | 207s |
| **State-of-the-art Results** | | | | | | | |
| BiDAF (Seo et al., 2016b) | - | - | - | 81.0 | 87.4 | - | - |
| R-net (Wang et al., 2017) | - | - | - | **82.1** | **88.1** | - | - |

Table 3: Exact match (EM) and F1 scores of various models on SQuAD (Section 4.2). We also report the total processing time per epoch and the time spent in RNN computations. SRU outperforms the LSTM models, and is more than six times faster than `cuDNN LSTM`. We also list the state-of-the-art test results for the EM and F1 metrics as listed on the leaderboard on December, 2017. Both state-of-the-art methods use RNNs, and can potentially benefit from our approach.

## 4.2 QUESTION ANSWERING

**Dataset** We use the Stanford Question Answering Dataset (SQuAD; Rajpurkar et al., 2016). SQuAD is one of the largest machine comprehension datasets, and includes over 100K question-answer pairs extracted from Wikipedia articles. We use the standard train and development sets.

**Setup** We experiment with the Document Reader model (Chen et al., 2017), and compare variants that use LSTM, as in the original setup, and SRU. We use the open source re-implementation.[3] Due to minor differences, this version performs 1% worse compared to the reported results when using the same training options. Following the author suggestions, we use a learning rate of 0.001 instead of 0.002, the `Adamax` (Kingma & Ba, 2014) optimizer, and separately tuned dropout rates for the RNN and word embeddings. This gives results comparable to the original paper. All models are trained for up to 50 epochs, batch size 32, a fixed learning rate of 0.001, and hidden dimensionality of 128. We use a dropout of 0.5 for input word embeddings, 0.2 for SRU layers, and 0.3 for LSTM layers.

**Results** Table 3 summarizes our results and state-of-the-art results as of December, 2017. LSTM models achieve 69.6% exact match and 78.9% F1 score. These results are comparable to the original work (Chen et al., 2017). SRU models achieve 70.3% exact match and 79.5% F1 score, outperforming the LSTM models. Moreover, SRU exhibits 6x to 10x speed-up, more than 69% reduction in total training time. The most recent state-of-the-art methods both use RNNs to encode text, and can potentially benefit from our approach.

## 4.3 LANGUAGE MODELING

**Dataset** We use the Penn Treebank corpus (PTB). The processed data and splits are taken from Mikolov et al. (2010). The data contains about 1M tokens with a truncated vocabulary of 10k. Following standard practice, the training data is treated as a long sequence split to mini batches, the models are trained using truncated back-propagation-through-time (BPTT; Williams & Peng, 1990).

**Setup** We largely follow the configuration of prior work (Zaremba et al., 2014; Gal & Ghahramani, 2016; Zoph & Le, 2016). We use a batch size of 32 and truncated back-propagation with 35 steps. The dropout probability is 0.75 for the input embedding and output softmax layer. The standard dropout and variational dropout probability are 0.2 for stacked RNN layers. We use SGD with an initial learning rate of 1.0 and gradient clipping. We train up to 300 epochs, and start to decrease the learning rate by a factor of 0.98 after 175 epochs. We use the identity activation function for $g(\cdot)$.

---

[3] https://github.com/hitvoice/DrQA

| Model | # layers | Size | Dev | Test | Time per epoch | |
| | | | | | RNN | Total |
| --- | --- | --- | --- | --- | --- | --- |
| LSTM (Zaremba et al., 2014) | 2 | 66m | 82.2 | 78.4 | | |
| LSTM (Press & Wolf, 2017) | 2 | 51m | 75.8 | 73.2 | | |
| LSTM (Inan et al., 2016) | 2 | 28m | 72.5 | 69.0 | | |
| RHN (Zilly et al., 2017) | 10 | 23m | 67.9 | 65.4 | | |
| KNN (Lei et al., 2017) | 4 | 20m | - | 63.8 | | |
| NAS (Zoph & Le, 2016) | - | 25m | - | 64.0 | | |
| NAS (Zoph & Le, 2016) | - | 54m | - | 62.4 | | |
| cuDNN LSTM | 2 | 24m | 73.3 | 71.4 | 53s | 73s |
| cuDNN LSTM | 3 | 24m | 78.8 | 76.2 | 64s | 79s |
| SRU | 3 | 24m | 68.0 | 64.7 | 21s | 44s |
| SRU | 4 | 24m | 65.8 | 62.5 | 23s | 44s |
| SRU | 5 | 24m | 63.9 | 61.0 | 27s | 46s |
| SRU | 6 | 24m | 63.4 | 60.3 | 28s | 47s |

Table 4: Language modeling perplexities on the PTB dataset. Models in comparison are trained using similar regularization and learning strategy: variational dropout is used except for (Zaremba et al., 2014), (Press & Wolf, 2017) and cuDNN LSTM; input and output word embeddings are tied except for (Zaremba et al., 2014); SGD with learning rate decay is used for all models.(Section 4.3). We also report time per training epoch, including for the entire architecture (Total) and for the RNN only.

**Results** Table 4 shows perplexity results. We use a parameter budget of 24 million for a fair comparison. cuDNN LSTM obtains a perplexity of 71.4 with 73-79 seconds per epoch. This result is worse than most prior work. We attribute this difference to the lack of variational dropout support in the cuDNN implementation. SRU obtains better perplexity compared to cuDNN LSTM and prior work, reaching 64.7 with three recurrent layers and 60.3 with six layers.[4] SRU also provides better speed-perplexity trade-off: training a six-layer RNN takes 47 seconds per epoch.

## 4.4 MACHINE TRANSLATION

**Dataset** We use the WMT 2014 English→German translation task. We pre-process the training corpus following standard practice (Peitz et al., 2014; Li et al., 2014; Jean et al., 2015). About 4M translation pairs are left after processing. The news-test-2014 data is used as the test set and the concatenation of news-test-2012 and news-test-2013 is used as the development set.

**Setup** We use OpenNMT (Klein et al., 2017), and extend the Pytorch version[5] with our SRU implementation. OpenNMT uses a seq2seq model in a recurrent encoder-decoder architecture with attention (Luong et al., 2015). By default, the model provides $\mathbf{h}_{t-1}$, the hidden state of decoder at step $t-1$, as input to step $t$. Although this can potentially improve translation quality, it impedes parallelization and slows down training. We disable this option. All models are trained with hidden state and word embedding size of 500, 15 epochs, SGD with initial learning rate of 1.0, and batch size 64. We modify the default of OpenNMT, and use a dropout rate of 0.1 and a weight decay of $10^{-5}$. This leads to better results for both RNN implementations.

**Results** Table 5 shows the translation results. We obtain better BLEU scores compared to the reports OpenNMT results (Klein et al., 2017). SRU with 10 stacked layers achieves a BLEU score of 20.7 while cuDNN LSTM achieves 20.45 using more parameters and more training time. SRU scales better, and we can stack many layers of SRU without significant time increase. Each additional SRU layer in encoder and decoder adds four minutes per training epoch, while adding an LSTM layer adds 23 minutes. In comparison, the other operations (e.g., attention and output softmax) take about 95 minutes. We do not observe over-fitting on the development set, even when using 10 layers.

---

[4]Melis et al. (2017) recently demonstrated that LSTM models can achieve a perplexity of 58 via careful regularization and hyper-parameter tuning. We leave these optimizations for future work.

[5]https://github.com/OpenNMT/OpenNMT-py

| OpenNMT default setup | # layers | Size all | Size w/o Emb. | Test BLEU | Time in RNNs |
|---|---|---|---|---|---|
| Klein et al. (2017) | 2 | - | - | 17.60 | |
| Klein et al. (2017) + BPE | 2 | - | - | 19.34 | |
| cuDNN LSTM (wd = 0) | 2 | 85m | 10m | 18.04 | 149 min |
| cuDNN LSTM (wd = $10^{-5}$) | 2 | 85m | 10m | 19.99 | 149 min |
| **Our setup** | | | | | |
| cuDNN LSTM | 2 | 84m | 9m | 19.67 | 46 min |
| cuDNN LSTM | 3 | 88m | 13m | 19.85 | 69 min |
| cuDNN LSTM | 5 | 96m | 21m | 20.45 | 115 min |
| SRU | 3 | 81m | 6m | 18.89 | 12 min |
| SRU | 5 | 84m | 9m | 19.77 | 20 min |
| SRU | 6 | 85m | 10m | 20.17 | 24 min |
| SRU | 10 | 91m | 16m | 20.70 | 40 min |
| **State-of-the-art Results** | | | | | |
| GNMT Ensemble (Wu et al., 2016b) | 8 | - | | 26.3 | - |
| ConvS2S Ensemble (Gehring et al., 2017) | 20 | - | | 26.4 | - |
| Transformer (Vaswani et al., 2017) | 6 | | 64m | 27.3 | - |
| Transformer (Vaswani et al., 2017) | 6 | | 213m | **28.4** | - |

Table 5: English-German translation results (Section 4.4). We list the total number of parameters (Size all) and the number excluding word embeddings (Size w/o Emb.). Our setup disables $\mathbf{h}_{t-1}$ input, which significantly reduces the training time. Timings are performed on a single Nvidia Titan X Pascal GPU. We also list recent state-of-the-art results.

We also include the most recent state-of-the-art results in Table 5. In our experiments, we use the default model provided by OpenNMT as our baseline architecture. Both RNN and non-RNN architectures have achieved significantly better BLEU scores on the En-Ge translation task than the OpenNMT default model. Combining SRU with state-of-the-art architectures such as Transformer is an important research direction, which we plan to explore in future work. For instance, with SRU only 4 minutes are spent per recurrent layer. This reduction in RNN time enables introducing recurrent computations into the Transformer architecture.

## 4.5 SPEECH RECOGNITION

**Dataset**  We use the Switchboard-1 corpus (Godfrey et al., 1992). The training data includes about 300 hours of speech from 4,870 sides of conversations between 520 speakers. The test data includes about two hours of speech from 40 sides of conversations from the 2000 Hub5 evaluation.

**Setup**  We use Kaldi (Povey et al., 2011) for feature extraction, decoding, and training of initial HMM-GMM models. We use the standard Kaldi recipes to train maximum likelihood-criterion context-dependent speaker adapted acoustic models with Mel-Frequency Cepstral Coefficient (MFCC). We apply forced alignment to generate labels for neural network acoustic model training. We use the Computational Network Toolkit (CNTK; Yu et al., 2014) instead of PyTorch. We experiment with uni-directional and bi-directional models, with and without state-level Minimum Bayes Risk (sMBR) training (Kingsbury et al., 2012). We use a trigram based language model instead of a RNN based language model on top of the acoustic model. Word error rates are reported after 4-gram LM rescoring of lattices generated using a trigram LM as described in Povey et al. (2016). See Appendix B for the complete setup details.

**Results**  Table 6 summarizes the results. CNTK uses a special batching algorithms for RNNs, and hence we were not able to use our customized SRU kernel. However, even without any kernel optimization, the SRU is faster than an LSTM using the same number of parameters. SRU also achieves better results on WER. Adding the highway connections Srivastava et al. (2015) to the LSTM performs slightly worse than the baseline. Removing the dependency on the internal state $\mathbf{h}$ in

| Model | # layers | # Parameters | WER | Time per epoch |
|---|---|---|---|---|
| LSTM | 5 | 47M | 11.9 | 136 min |
| LSTM + Seq | 5 | 47M | 10.8 | - |
| Bi-LSTM | 5 | 60M | 11.2 | 273 min |
| Bi-LSTM + Seq | 5 | 60M | 10.4 | - |
| LSTM with highway (remove **h**) | 12 | 56M | 12.5 | 210 min |
| LSTM with highway | 12 | 56M | 12.2 | 296 min |
| SRU | 12 | 56M | 11.6 | 113 min |
| SRU + sMBR | 12 | 56M | 10.0 | - |
| Bi-SRU | 12 | 74M | 10.5 | 220 min |
| Bi-SRU + sMBR | 12 | 74M | **9.5** | - |
| Very Deep CNN + sMBR  (Saon et al., 2016) | 10 | | 10.5 | - |
| LSTM + LF-MMI  (Povey et al., 2016) | 3 | | 10.3 | - |
| Bi-LSTM + LF-MMI  (Povey et al., 2016) | 3 | | 9.6 | - |

Table 6: Word error rate (WER) for speech recognition (Section 4.5). The timing numbers are based on a naive implementation of SRU in CNTK. No CUDA-level optimizations are performed.

the LSTM can improve the speed but causes a slight decrease in performance. Appendix C includes further experiments with different highway structures and number of layers.

## 5 CONCLUSION

We present Simple Recurrent Unit (SRU), a recurrent architecture that is as fast as CNN and easily scales to over 10 layers. Our evaluation on a variety of NLP and speech recognition tasks demonstrates the effectiveness of SRU. We open source our implementation to facilitate future research.

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

## A    COMPARISON OF MODEL VARIANTS AND QUASI-RNN

SRU and Quasi-RNN (Bradbury et al., 2017) both were developed with the goal of speeding up the computation of recurrent neural networks. The Quasi-RNN design aims to combine $k$-gram convolutions with adaptive pooling (i.e., `fo-pooling`) instead of traditional order-oblivious pooling, such as max pooling or average pooling. Similar to other convolutional architectures (Kalchbrenner et al., 2014; Kim, 2014; Wang et al., 2015; Dauphin et al., 2017), a $k$-gram filter width $> 1$ is used throughout the experiments reported.

While Quasi-RNN is based on adding light recurrence to convolutional network, SRU is an instance of the recurrent architectures introduced by Lei et al. (2017). While this prior work focused on theoretical characteristics of the networks and their generalization, we focus on the implementation details of SRU, including practical optimizations, and on its applicability to a wide range of tasks. The SRU computation is similar to the degenerate case of setting $k = 1$ in the $k$-gram filter in Quasi-RNN. The computation is then reduced to a simple feed-forward transformation (matrix multiplication), and loses the properties and quality of a convolution.

The SRU architecture and results we present also differ from Quasi-RNN in key technical decisions:

- **CUDA Optimization:** While the Quasi-RNN implementation of `fo-pooling` is done as a CUDA kernel function. The rest of the element-wise computation is performed via separate function calls on top of the software library (e.g., Chainer). This choice does not allow this type of architecture to achieve its full potential. In contrast, we implement element-wise fusion to enable further speed optimization (Section 2.3).

- **Activation and Highway Connections:** SRU and Quasi-RNN differ in how the non-linear activation function is applied. Quasi-RNN follows the common practice with convolutional models, where the non-linear activation is applied with the convolution operation before pooling (i.e., $\tilde{\mathbf{x}} = \tanh(\mathbf{W} * \mathbf{x})$). In contrast, in SRU the activation function $g(\cdot)$ is applied to compute the internal state (i.e., $\mathbf{h}_t = g(\mathbf{c}_t)$) following the derivation in (Lei et al., 2017). SRU also includes highway connections in the architecture for better generalization of deep networks.

**Effect of Element-wise Fusion**    We compare the speed of the `fo-pooling` implementation used in Quasi-RNN and the fused kernel implementation used in SRU. We implement a version of uni-directional and bi-directional SRU that uses `fo-pooling` and separate element-wise function calls in PyTorch. Figure 3 (left) presents the speed comparison on five classification tasks (Section 4.1) and SQuAD (Section 4.2). The fused element-wise kernel achieves 24% to 94% speed improvement across the six benchmarks.

**Effect of Activation and Highway Connection**    We compare the performance of SRU variants with different activation functions and Quasi-RNN with filter width $k = 1$ on the classification and SQuAD datasets. For the classification tasks, we train using using the `Adam` optimizer with default 0.001 learning rate, 0 weight decay, and dropout probability tuned from values $\{0.1, 0.3, 0.5, 0.7\}$. We perform three trials for 10-fold cross validation for each model and dataset. We report the average test accuracy of model configurations that achieve the best development results. For SQuAD, we train all models for a maximum of 100 epochs using the `Admax` optimizer with learning rate 0.001. We perform three independent trials, and report the average performance. Figure 3 (right), Table 4, and Table 5 summarize the results:

- SRU performs at least as good as Quasi-RNN, and often outperforms it. This illustrates the advantages of the design choices in SRU, including the highway connections and activation implementation.

- We observe that the optimal choice of activation function varies depending on the task. For example, we found ReLU activation to work best on the classification benchmarks, but the identity activation performs the best on SQuAD, while ReLU performs worst.

- The effect of highway connections is best highlighted in the question answering dataset. Without the highway connections, we observe a performance decrease when stacking more than four recurrent layers (Table 5). In contrast, adding highway connections results in $> 1\%$ absolute improvement of exact match score. Figure 3 (right) also shows that no

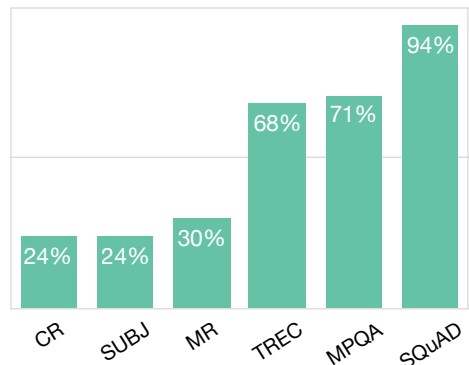 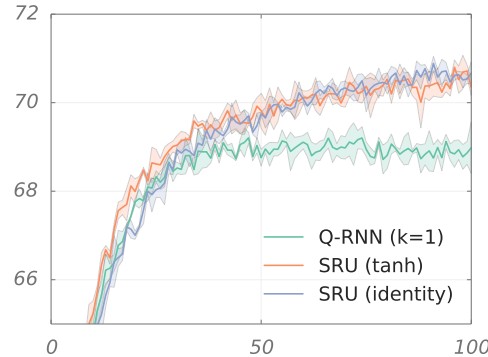

Figure 3: Left: relative speed improvement of fused kernel (SRU) over `fo-pooling` kernel (Quasi-RNN) on various benchmarks. Timings are performed on a desktop machine with GeForce GTX 1070 and Intel Core i7-7700K Processor. Right: mean exact match (EM) score of 5-layer SRU and Quasi-RNN on SQuAD as a function of the number of epochs. Models are trained for a maximum of 100 epochs using `Admax` optimizer with learning rate 0.001.

| Model | CR | SUBJ | MR | TREC | MPQA |
|---|---|---|---|---|---|
| Quasi-RNN ($k = 1$) | 83.6 ±2.0 | 93.3 ±0.8 | 81.6 ±1.1 | 92.7 ±0.6 | 89.6 ±1.2 |
| Quasi-RNN ($k = 1$) + highway | 84.0 ±1.9 | 93.4 ±0.8 | 82.1 ±1.2 | 93.2 ±0.6 | 89.6 ±1.2 |
| SRU (identity) | 84.1 ±1.9 | 93.5 ±0.7 | 82.1 ±1.0 | 93.8 ±0.4 | 89.7 ±1.1 |
| SRU (tanh) | 84.2 ±1.7 | 93.5 ±0.8 | 82.1 ±1.1 | 93.9 ±0.6 | 89.8 ±1.0 |
| SRU (ReLU) | 84.7 ±1.9 | 93.7 ±0.9 | 82.5 ±1.1 | 93.7 ±0.5 | 89.8 ±1.0 |

Figure 4: Comparison between Quasi-RNN and SRU on classification benchmarks. We perform 3 independent trials of 10-fold cross validation ($3 \times 10$ runs) for each model and dataset. We report the average test accuracy of model configurations that achieve the best dev result. All models are trained using `Adam` optimizer with default learning rate $= 0.001$, weight decay $= 0$ and dropout probability tuned from values $\{0.1, 0.3, 0.5, 0.7\}$.

| Model | Number of recurrent layers | | |
|---|---|---|---|
| | 4 | 5 | 6 |
| Quasi-RNN ($k = 1$) | 70.0 ±0.2 | 69.5 ±0.2 | 69.3 ±0.1 |
| Quasi-RNN ($k = 1$) + highway | 70.1 ±0.1 | 70.7 ±0.2 | 70.6 ±0.1 |
| SRU (ReLU) | 70.0 ±0.2 | 70.1 ±0.1 | 70.4 ±0.2 |
| SRU (tanh) | 70.3 ±0.1 | 70.9 ±0.2 | 70.7 ±0.2 |
| SRU (identity) | 70.5 ±0.2 | 71.0 ±0.1 | 71.1 ±0.1 |

Figure 5: Comparison between Quasi-RNN and SRU on the SQuAD benchmark. We perform 3 independent trials with a maximum of 100 training epochs. We report the average exact match (EM) score of each model configuration. Models are trained using `Admax` with learning rate 0.001.

over-fitting occurs on the development set within $100$ training epochs. This suggests that models with highway connections are likely to generalize better.

## B  SPEECH RECOGNITION EXPERIMENTAL SETUP DETAILS

Following Sainath et al. (2015), all weights are randomly initialized from the uniform distribution with range $[-0.05, 0.05]$, and all biases are initialized to $0$ without generative or discriminative pre-training (Seide et al., 2011). All neural network models, unless noted otherwise, are trained with a cross-entropy criterion using truncated BPTT. No momentum is used for the first epoch, and a momentum of $0.9$ is used for subsequent epochs (Zhang et al., 2015). We apply an $L2$ constraint regularization with weight $10^{-5}$ (Hinton et al., 2012) .

We experiment with uni-directional and bi-directional models. To train the uni-directional model, we unroll 20 frames and use 80 utterances in each mini-batch. We also delayed the output of the LSTM and SRU by 10 frames as suggested by Sak et al. (2014) to add more context. To train the bidirectional model, we use the latency-controlled method described in Zhang et al. (2015). We set $N_c = 80$ and $N_r = 20$ and processed 40 utterances simultaneously. In addition to our vanilla model, we also experiment state-level Minimum Bayes Risk (sMBR) training (Kingsbury et al., 2012). To train the recurrent model with the sMBR criterion, we adopted the two-forward-pass method described by Zhang et al. (2015), and processed 40 utterances simultaneously.

The input features for all models are 80-dimensional log Mel filterbank features computed every ten milliseconds, with an additional 3-dimensional pitch features. The output targets are 8802-context-dependent triphone states, of which the numbers are determined by the last HMM-GMM training stage.

Our setup can potentially improve if we incorporate some recent techniques that are applicable to SRU. For example, LF-MMI for sequence training, i-vectors for speaker adaptation, and speaker perturbation for data augmentation were applied by Povey et al. (2016). All of these techniques can also been used for SRU. Moreover, different highway variants such as grid LSTM (Hsu et al., 2016) can also further boost our model.

## C  ADDITIONAL SPEECH RECOGNITION ANALYSIS

**Baseline**    To identify the LSTM baseline used in Section 4.5, we experiment with varying the number of layers and parameters. Table 7 shows the performance of different settings. We follow the setup of Sak et al. (2014).[6] The best LSTM baseline is using five layers with $1024$ units in each layer.

| Model | # layers | # Parameters | WER |
|---|---|---|---|
| LSTM with projection  (Sak et al., 2014) | 5 | 28M | 12.2 |
| LSTM | 3 | 30M | 12.5 |
| LSTM (S) | 5 | 28M | 12.5 |
| LSTM | 5 | 47M | 11.9 |
| LSTM (L) | 5 | 94M | 12.0 |
| LSTM | 6 | 56M | 12.3 |

Table 7: Word error rate (WER) for LSTM baselines on the Switchboard-1 corpus (Section 4.5). LSTM has 1024 cells for each layer, LSTM (S) has 750 cells for each layer, and LSTM (L) has 1560 cells for each layer. LSTM with projection contains 1024 cells and a 512-node linear projection layer is added on top of each layer output.

**Effect of Highway Transform for SRU**    The dimensionality of the input $\mathbf{x}_t$ and $\mathbf{h}_t$ must be equal in the computation of $\mathbf{h}_t$ (Equation 5). However, when this is not the case, for example as in the first layer of the SRU, we can use a linear projection $\mathbf{W}_h^l$ to match the dimensions at layer $l$. The

---

[6]We do not a projection layer as we found the vanilla LSTM model performs better (Table 7).

modified version of Equation 5 will then be:

$$\mathbf{h}_t = \mathbf{r}_t \odot g(\mathbf{c}_t) + (1 - \mathbf{r}_t) \odot \mathbf{W}_h^l \mathbf{x}_t \ \ .$$

We can also use a square matrix $\mathbf{W}_h^l$ for every layer. Table 8 shows that adding this transformation significantly reduces the word error rate from $12.6\%$ to $11.8\%$ when using the same number of parameters. This transformation is outside the recurrent loop, and can be parallelized to be computed efficiently.

| Model | # layers | # Parameters | WER |
|---|---|---|---|
| SRU (no $\mathbf{W}_h^l, l > 1$) | 16 | 56M | 12.6 |
| SRU | 12 | 56M | 11.8 |

Table 8: Word error rate (WER) comparison of the effect of transformation in the highway connection. SRU (no $\mathbf{W}_h^l, l > 1$) includes the transform in the first layer only to align the dimensionality. The second line includes the transformation for every layer.

**Effect of Depth for SRU**   We study the effect of SRU depth on performance.[7] Table 9 shows word error rate for SRU with different number of layers. The SRU model outperforms the LSTM model with 10 layers and the same number of parameters, while provide a 1.4x speed-up, even though it is using a non-optimized implementation in CNTK.[8] The speed gains are mainly a result of requiring less matrix multiplications. The best performance is achieved with 12 layers.

| Model | # layers | # Parameters | WER | Time per epoch |
|---|---|---|---|---|
| LSTM | 5 | 47M | 11.9 | 136 min |
| SRU | 10 | 47M | 11.8 | 97 min |
| SRU | 12 | 56M | 11.5 | 113 min |
| SRU | 16 | 72M | 11.5 | 146 min |
| SRU | 20 | 89M | 11.8 | 170 min |

Table 9: Word error rate (WER) and time per training epoch as function of SRU depth for speech recognition (Section 4.5). The timing numbers are based on a naive implementation of SRU in CNTK. No CUDA-level optimizations are performed.

---

[7]In initial experiments, we observed that depth is more important for performance than width. Therefore, we focus our experiments on the effect of network depth. We leave more conclusive experiments about the trade-offs between depth and width for future work.

[8]The CNTK implementation does not use the customized SRU kernel.

