# OpenReview forum: "Training RNNs as Fast as CNNs"
_ICLR.cc/2018/Conference — Reject_

### Official Review · AnonReviewer2 · 2017-11-21
**Nice idea, tested extensively**

**Rating:** 7
**Confidence:** 4

**Review:**

This work presents the Simple Recurrent Unit architecture which allows more parallelism than the LSTM architecture while maintaining high performance.

Significance, Quality and clarity:
The idea is well motivated: Faster training is important for rapid experimentation, and altering the RNN cell so it can be paralleled makes sense.
The idea is well explained and the experiments convince that the new architecture is indeed much faster yet performs very well.

A few constructive comments:
- The experiment’s tables alternate between “time” and “speed”, It will be good to just have one of them.
- Table 4 has time/epoch yet only time is stated

---

> ### Author Response · Authors · 2017-12-10
> **Review response**
>
> Thank you for the comments and feedback.
>
> We agree that having both “time” and “speed” in the tables are confusing. “Time/epoch” in Table 4 is misleading. We will use “Time per epoch” or simply “Time” instead.
>
> We will address your feedback in the next version. Thanks!

---

> ### Author Response · Authors · 2017-12-30
> **Tables clarified**
>
> The latest revision contains fixes to the tables and unifies the measurements used. Thanks for the suggestion

---

### Official Review · AnonReviewer3 · 2017-11-27
**Very useful RNN cell with ok results but over-hyped presentation.**

**Rating:** 8
**Confidence:** 5

**Review:**

The authors introduce SRU, the Simple Recurrent Unit that can be used as a substitute for LSTM or GRU cells in RNNs. SRU is much more parallel than the standard LSTM or GRU, so it trains much faster: almost as fast as a convolutional layer with properly optimized CUDA code. Authors perform experiments on numerous tasks showing that SRU performs on par with LSTMs, but the baselines for these tasks are a little problematic (see below).

On the positive side, the paper is very clear and well-written, the SRU is a superbly elegant architecture with a fair bit of originality in its structure, and the results show that it could be a significant contribution to the field as it can probably replace LSTMs in most cases but yield fast training. On the negative side, the authors present the results without fully referencing and acknowledging state-of-the-art. Some of this has been pointed out in the comments below already. As another example: Table 5 that presents results for English-German WMT translation only compares to OpenNMT setups with maximum BLEU about 21. But already a long time ago Wu et. al. presented LSTMs reaching 25 BLEU and current SOTA is above 28 with training time much faster than those early models (https://arxiv.org/abs/1706.03762). While the latest are non-RNN architectures, a table like Table 5 should include them too, for a fair presentation. In conclusion: the authors seem to avoid discussing the problem that current non-RNN architectures  could be both faster and yield better results on some of the studied problems. That's bad presentation of related work and should be improved in the next versions (at which point this reviewer is willing to revise the score). But in all cases, this is a significant contribution to deep learning and deserves acceptance.

Update: the revised version of the paper addresses all my concerns and the comments show new evidence of potential applications, so I'm increasing my score.

---

> ### Author Response · Authors · 2017-12-10
> **Review response**
>
> Thank you for the comments and feedback.
>
> == Paper revision ==
> We will include missing SOTA results and related work for translation as pointed by R3, as we already included for language modeling and speech. We will update the table in the next version.
>
> == Clarification on our experiments ==
> The goal of our experiments is not to outperform previous SOTA. Instead, the experiments were designed to study SRU’s effectiveness on a broad set of realistic applications via fair comparison. Therefore, we emphasized using existing open source implementations for MT and QA. Different implementations (network architectures, data processing etc.) have non-trivial impact on the final numbers. To the best of our effort, we aimed to avoid this influencing our experiments. Therefore, in the current version, Tables 1, 3, and 5 only compare the results of using LSTM / SRU / Conv2d as building blocks in existing models such DrQA and OpenNMT. We definitely agree that including SOTA models in these tables will improve our presentation. Thank you for the suggestion.
>
> == Non-RNN architectures ==
> Thank you for the comment. We will include discussions of non-RNN architectures. Our contribution is orthogonal to recent architectures, such as Transformer (https://arxiv.org/abs/1706.03762), which is a novel combination of multi-head attention and feed-forward networks. Part of the motivation behind the Transformer architecture is the computational bottleneck of recurrent architectures. With SRU this is not longer the case. In fact, we observe in the translation model that only 4 minutes are spent per SRU layer, and 96 minutes are spent in the attention+softmax computation. An interesting direction for future work is combining the SRU and Transformer architectures to gain the benefits of both. While this is an important problem, it is beyond the scope of our experiments.

---

> > ### Comment · AnonReviewer3 · 2017-12-23
> > **Still no SOTA results in the tables?**
> >
> > I am not sure how to interpret the comment about the Transformer architecture. There is a table with results in your paper that are far below SOTA and it doesn't even mention this -- it looks like clearly misleading presentation, and with your comment it starts looking like it's misleading on purpose. Thus I'm lowering my score until the presentation is improved. In particular, your results are below 21 BLEU which is very far apart from the 28 BLEU of the Transformer -- the suggestion you make in the comment (that architectures like Transformer may not be needed with SRUs) seems to be far from conclusive at this point. Please present your work fairly and compare to existing SOTA -- it's a very good work, but the presentation is misleading.

---

> > > ### Author Response · Authors · 2017-12-24
> > > **SOTA included**
> > >
> > > Hi,
> > >
> > > Sorry for the delayed revision. The state-of-the-art results have been included in the tables for both machine translation and reading comprehension tasks. We hope the results are now better presented.
> > >
> > > Please let use know if other related work should be included. We are happy to address additional comments.
> > >
> > > Also, we didn't mean that "Transformer may not be needed with SRUs". As discussed in the introduction of the Transformer paper, RNN is discarded in Transformer architecture due to the difficulty to parallelize recurrent computation. Thus, it is perhaps possible to "achieve the best of both worlds" by incorporating SRU into Transformer (e.g. substituting the FFN sub-unit).

---

> > > > ### Comment · AnonReviewer3 · 2017-12-24
> > > > **Looks much better, thank you!**
> > > >
> > > > Thank you for the new version of the paper. It looks much better, and I misunderstood the comments about Transformer. Indeed, combining it with SRUs could bring the best of both worlds and improve results even more. I have no more objections to accepting this work and I see its big potential, adjusting my review.

---

### Official Review · AnonReviewer1 · 2017-12-08
**Low novelty and lacks comparison with obvious baselines**

**Rating:** 4
**Confidence:** 5

**Review:**

The authors propose to drop the recurrent state-to-gates connections from RNNs to speed up the model. The recurrent connections however are core to an RNN. Without them, the RNN defaults simply to a CNN with gated incremental pooling. This results in a somewhat unfortunate naming (simple *recurrent* unit), but most importantly makes a comparison with autoregressive sequence CNNs [ Bytenet (Kalchbrenner et al 2016), Conv Seq2Seq (Dauphin et al, 2017) ] crucial in order to show that gated incremental pooling is beneficial over a simple CNN architecture baseline.

In essence, the paper shows that autoregressive CNNs with gated incremental pooling perform comparably to RNNs on a number of tasks while being faster to compute. Since it is already extensively known that autoregressive CNNs and attentional models can achieve this, the *CNN* part of the paper cannot be counted as a novel contribution. What is left is the gated incremental pooling operation; but to show that this operation is beneficial when added to autoregressive CNNs, a thorough comparison with an autoregressive CNN baseline is necessary.

Pros:
- Fairly well presented
- Wide range of experiments, despite underwhelming absolute results

Cons:
- Quasi-RNNs are almost identical and already have results on small-scale tasks.
- Slightly unfortunate naming that does not account for autoregressive CNNs
- Lack of comparison with autoregressive CNN baselines, which signals a major conceptual error in the paper.
- I would suggest to focus on a small set of tasks and show that the model achieves very good or SOTA performance on them, instead of focussing on many tasks with just relative improvements over the RNN baseline.

I recommend showing exhaustively and experimentally that gated incremental pooling can be helpful for autoregressive CNNs on sequence tasks (MT, LM and ASR). I will adjust my score accordingly if the experiments are presented.

---

> ### Author Response · Authors · 2017-12-10
> **Review response**
>
> Thank you for the comments and feedback. We respond to the concerns and questions raised in three section.
>
> == Recurrent or convolution ==
> We wish to certain aspects pertaining to the distinction between recurrent and convolution architectures as we use in the paper:
>
> (1) SRU only applies simple matrix multiplications (Wx_t) for each x_t. This is not a typical convolution operation that is applied over k consecutive tokens. While matrix multiplication can be considered a convolution operation of k=1, this entails that feed-forward networks (FFN) are also a convolutional network. More important, with k=1 there is no convolution over the words, which is the key aim of CNNs for text processing, for example to reason about n-gram patterns. Therefore, while notationaly correct, we consider the k=1 case to empty the term convolution from the meaning it is intended to convey, and do not use it in this way in the paper. That said, we discuss the relationship of these two types of computations in Appendix A, and will be happy to clarify it further in the body of the paper.
>
> (2) This being said, the effectiveness of SRU comes from the recurrent computation of its internal state c[t] (rather than applying conv operations). This internal state computation (referred to in the review as gated incremental pooling) is commonly used as the key component in gated RNN variants, including LSTM, GRU, RAN, MGU, etc.
>
> (3) Beyond the choice of terms, and even if we were to consider SRU as a special type of CNN (with k=1), to the best of our knowledge, our study is the first to demonstrate that k=1 suffices to work effectively across a range of NLP and speech tasks. This emphasis on efficiency goes beyond prior work (e.g. Bytenet, ConvS2S and Quasi-RNN), where conv operations of k=3,4,etc are used throughout the experiments. This allows us to simplify architecture tuning and significantly speeds up the network, which is the main focus of this work. As shown in Figure 2, SRU operates faster than a single conv operation of k=3.
>
> (4) Quasi-RNN, T-RNN and T-LSTM (https://arxiv.org/pdf/1602.02218.pdf) have also used “RNN” in naming, despite defaulting to CNN with gated incremental pooling. Broadly speaking, we consider any unit that successively updates state c[t] based on current input x[t] and the previous vector c[t-1] (as a function c[t]=f(x[t], c[t-1])) as a recurrent unit. We will clarify this better in the paper.
>
> == Quasi-RNN and scale of tasks ==
> We discuss the comparison to Quasi-RNN in Appendix A, and emphasize the critical differences. In our experiments, the training time of a single run on machine translation takes about 2 days, and 4 days on speech on a Titan X GPU.
>
> == Wide experiments vs deep experiments ==
> Our experiments are aimed to study SRU’s effectiveness on a broad set of realistic applications via fair comparison. We discuss this more in our response to Reviewer 3.
>
> Our work focuses on practical simplifications, optimizations, and the applicability of SRU to a wide range of realistic tasks. Although we do not perform an exhaustive hyper-parameter / architecture tuning on each task given space and time constraints, we do see an improvement over deep CNNs on speech recognition. Similar results have been reported in prior work such as RCNN (Lei et al; 15,16), KNN (Lei et al; 17) and Quasi-RNN (Bradbury et al; 17), demonstrating that gated pooling is helpful for CNN-type models on tasks such as classification, retrieval, LM etc.

---

### Public Comment · (anonymous) · 2017-11-09
**Good paper**

If the original result (arxiv) was already pretty surprising, this result seems to be even better? It seems a solid 3x speed-up is expected, and it can train a crazy number of layers (10 layers in MT).

In the actual code on github, it says "use_tanh=0" and set highway bias to "-3". These intuitions are not explained in the paper. Can the author offer some understanding into them? It seems that identity is better than tanh in the appendix...but then again...some explanation?

---

> ### Author Response · Authors · 2017-11-11
> **activation and highway bias**
>
> Thank you for the comment.
>
> The identity activation (use_tanh=0) and non-zero highway bias are applied only on language modeling following a few of recent papers such as
>   - language modeling via gated convolutional network: https://arxiv.org/pdf/1612.08083.pdf
>   - recurrent highway network: https://arxiv.org/abs/1607.03474
>
> We expect the model to perform better on other tasks as well by initializing a non-zero highway bias, since it can help to balance gradient propagation and model complexity (non-linearity) from layer stacking. This is recommended in the original highway network paper (https://arxiv.org/abs/1505.00387). However, we choose to use zero highway bias on other tasks for simplicity.
>
> Regarding the choice of activation function:
>   - this could be an empirical question since the best activation varies across tasks / datasets (Appendix A)
>   - identity already works since the pre-activation state (i.e. c[t]) readily encapsulates sequence similarity computation. see the discussed related work (Lei et al 2017; section 2.1 & 2.2) https://arxiv.org/pdf/1705.09037.pdf
>
> Thank you again for bringing up the questions.

---

### Public Comment · ~Dmitriy_Serdyuk1 · 2017-11-15
**This is not SOTA on Switchboard**

You cannot claim state of the art on Switchboard. https://arxiv.org/pdf/1610.05256.pdf showed 7.7% WER (Table 8, first row). Unless you are using no LM here (you need to describe LM you used), you don't have SOTA.

Second, 300h training set is just not very interesting for current research on ASR, therefore not many paper publish results on it. Have you run your model on 2000h set?

---

> ### Author Response · Authors · 2017-11-15
> **Results on Switchboard**
>
> Thanks for your comments.
>
> Sorry for the confusing, we didn't use RNN-LM here (only N-gram). So the number we should compare with is 10.0 in Table 8. I think JHU recently have better number using the same language model with lattice-free MMI training. We will try this new loss later. But similar to RNN-LM, this is orthogonal to this paper, we are trying to compare with LSTM only for acoustic modeling.
>
> We haven't try it on 2000hrs. (1) To my understanding, there still lots of institute use 300hrs setup especially at school. If you check last year ICASSP, there are still many paper use 300hrs set, e.g. http://danielpovey.com/files/2017_spl_tdnnlstm.pdf. (2) In my experiences, 20000hrs vs. 300hrs do make a difference, especially for end-to-end system. But 2000hrs set and 300hrs usually don't have significant difference in term of testing the trend of the model quality (especially for HMM-NN hybrid system, model A > model B for 300hrs usually also hold for the full fisher set). Also, 300hrs usually take 4 days on a single GPU which is a reasonable setup for reproduce results.

---

> > ### Public Comment · ~Dmitriy_Serdyuk1 · 2017-11-15
> > **Results on Switchboard**
> >
> > Thanks for a quick response.
> >
> > Ok, I see now. Maybe you need to describe the setup more thoroughly and state which work you are basing your experiments on.

---

> > > ### Author Response · Authors · 2017-11-15
> > > **Results on Switchboard**
> > >
> > > Thank you! We will update it.

---

### Comment · AnonReviewer2 · 2017-11-15
**Nice work**

In Tables 6 / 9:
It is not clear why SRU model capacity was increased in depth (to 12 layers) and not in width, which would give an even faster model I would think. As you mention for LSTM 5 layers appear to be optimal, so it is surprising that 12 were needed for SRU.

---

> ### Author Response · Authors · 2017-11-15
> **Depth vs. width**
>
> In general we found that increasing depth is more helpful than increasing width as long as the width is in a reasonable size. I think this is because we drop "the dependency" between "h" and this context needs to be recovered by adding more layers. But since SWB training takes about 4 days, we didn't try all the configuration. That's why we didn't draw a conclusion on depth vs. width.

---

### Public Comment · (anonymous) · 2017-12-02
**Recall simple recurrent network**

Nice work! But isn't the name simple recurrent unit (SRU) a bit similar to the classic name "simple recurrent network" which often refers to both Jordan & Elman networks.
https://en.wikipedia.org/wiki/Recurrent_neural_network

---

### Public Comment · ~rajveer_gandhi1 · 2017-12-16

We are students at McGill University and were reviewing your paper, here are some of our results.

To reproduce these results, we created a Google Cloud Instance with similar hardware specifications. The authors performed their experiments on a desktop machine with a single NVIDIA GeForce GTX 1070 GPU, Intel Core i7-7700K Processor, using CUDA 8 and cuDNN 6021. Our cloud instance runs on a Haswell-based Intel x86_64 and uses NVIDIA's Tesla K80 GPU. We use the source code provided by the authors with minimal changes on our environment https://github.com/taolei87/sru to sucessfully reproduce the classification, question answering, and language modeling tasks. Our software environment consisted of the following packages: Ubuntu 16.04, CUDA 9.0.176, CuDNN 6.0, PyTorch 0.2.0.post4, Pynvrtc 8.0, and CuPY 4.0.0b1. Python 2.7 was used for all models except question answering, which required Python 3 due to the DrQA dependency.

For the classification task, we were able to reproduce their results on all six datasets. We trained SRU- and LSTM-based RNNs and a CNN on each dataset 5 times for 100 epochs each. In all instances, the SRU outperformed CNN and LSTM-based RNNs in terms of accuracy and overall training time. We observed  similar training times for all tests except MPQA and SST, where we observed wall clock training times nearly twice as long as reported. This could be explained by using 4 cores and a shared cloud GPU, where the authors had an 8-core CPU and dedicated GPU.

For the question answering model, we used an open source reimplentation of the Document Reader model https://github.com/hitvoice/DrQA with the suggested dropout rates. Despite our best efforts, we were not able to achieve the authors' reported baseline accuracy. We obtained an F1 score of 75.4 and 66% exact match for the LSTM-based RNN, which is 3% lower than reported. However when we trained the SRU model, we were able to obtain closer results to authors: 77.8 F1 score and 67.9 % exact match. This is within 1.5% of the reported results for SRU based training. Moreover, we observed 71% faster overall training when compared to the LSTM-based model, which aligns with the authors observed 69% increase in their published experiment.

For the language model, the published code ran essentially unmodified, allowing us to reproduce the paper's experimental results to within aproximately 1% error of the reported perplexity and wall clock runtime, for both cuDNN LSTM and SRU configurations, confirming state-of-the-art model performance for the setup described in the paper. Our final performance on the author-recommended hyperparameter settings (6 layers deep, 910 units wide) achieved test perplexity of 60.66 and validation perplexity of 64.17 after 300 training epochs.

The speech recognition model was the most challenging to set up and build. Due to unforseen difficulty in replicating the software environment, we were unable to reproduce the experiment as described. The authors use a forked version of CNTK with custom modifications to compare the bidirectional SRU to a latency-controlled bidirectional LSTM. Despite the authors timely assitance, we were unable to build the fork as described. Our efforts to reproduce the speech experiment are documented here: https://github.com/taolei87/sru/issues/36 https://github.com/taolei87/sru/issues/36.

Overall, we feel the SRU architecture offers important advantages for parallelism and scaling, facilitating the training of recurrrent neural networks on larger datasets with commodity hardware. It achieves higher accuracy in the same number of epochs as traditional LSTM-based RNNs, using less wall clock time, and demonstrates RNN training and inference need not be as sequential as previously believed. This suggests further research into parallelizable architectures may unlock similar gains in speedup and performance. For a detailed summary of our experimental results, our full report is available at the following URL: https://github.com/msalihs/sru/blob/master/comp551_reproducibility_project_group_RMB.pdf

---

### Author Response · Authors · 2017-12-24
**Revision with updated result tables**

We updated the paper to include recent state-of-the-art results for the QA and translation tasks to avoid confusion about how the results should be interpreted. We thank AnonReviewer3 for suggesting this.

---

### Decision · Program_Chairs · 2018-01-29
**ICLR 2018 Conference Acceptance Decision**

**Decision:**

Reject

**Comment:**

The paper presents Simple Recurrent Unit, which is characterised by the lack of state-to-gates connections as used in conventional recurrent architectures. This allows for efficient implementation, and leads to results competitive with the recurrent baselines, as shown on several benchmarks.

The submission lacks novelty, as the proposed method is essentially a special case of Quasi-RNN [Bradbury et al.], published at ICLR 2017. The comparison in Appendix A confirms that, as well as similar results of SRU and Quasi-RNN in Figures 4 and 5. Quasi-RNN has already been demonstrated to be amenable to efficient implementation and perform on a par with the recurrent baselines, so this submission doesn’t add much to that.